# Catalytic hydrogenation of olefins by a multifunctional molybdenum-sulfur complex

Minghui Xue[1], Zhiqiang Peng[2], Keyan Tao[2], Jiong Jia[1], Datong Song [3] ✉, Chen-Ho Tung[1] & Wenguang Wang [1,2] ✉

Exploration of molybdenum complexes as homogeneous hydrogenation catalysts has garnered significant attention, but hydrogenation of unactivated olefins under mild conditions are scarce. Here, we report the synthesis of a molybdenum complex, $[Cp*Mo(Ph_2PC_6H_4S-CH=CH_2)(Py)]^+$ (**2**), which exhibits intriguing reactivity toward $C_2H_2$ and $H_2$ under ambient pressure. This vinylthioether complex showcases efficient catalytic activity in the hydrogenation of various aromatic and aliphatic alkenes, demonstrating a broad substrate scope without the need for any additives. The catalytic pathway involves an uncommon oxidative addition of $H_2$ to the cationic Mo(II) center, resulting in a Mo(IV) dihydride intermediate. Moreover, complex **2** also shows catalytic activity toward $C_2H_2$, leading to the production of polyacetylene and the extension of the vinylthioether ligand into a pendant triene chain.

The multifaceted roles of metal sulfides in enzymes have generated sustained interest in exploring metal-sulfur complexes for the activation of small molecules[1–3]. Molybdenum-sulfur (Mo-S) is a crucial unit that constitutes the active sites of numerous metalloenzymes, including molybdenum nitrogenase, nitrate reductases, sulfite oxidase, and xanthine oxidoreductases[4–6]. With the aid of sulfur-based scaffolds, molybdenum modulates its flexible redox properties to complement the exceptional catalytic performances of enzymes[7,8]. Early studies on the synthetic Mo-S complexes such as $[CpMoSC_2H_4S]_2$[9] and $Cp_2Mo_2S_4$[10,11] revealed the diverse roles of sulfur in interacting with $C_2H_2$ and $H_2$ for bond cleavage and formations (Fig. 1)[12–15]. The intriguing reactivity of this family was illustrated by hydrogenation of the alkyne moiety in $[CpMoSC_2H_4S]_2$ (I) under mild conditions[9,10]. Also, the molybdenum-sulfur complexes have been found as efficient catalysts for electrochemical $H_2$ production[14] and N-heteroarene reduction[16]. These findings provide insights into exploring novel inexpensive metal catalysts for olefin hydrogenation[17].

Exploration of molybdenum complexes as homogeneous hydrogenation catalysts has garnered significant attention[18,19], and notable progress has been made in the ionic hydrogenation of C = O and C = N bonds[20–24]. Ionic hydrogenations were proposed to involve $H_2$ heterolytic cleavage to form critical metal-hydride (Mo−H) intermediate[18,22,23,25]. The groups of Bullock and Berke have demonstrated the utilization of bifunctional molybdenum complexes, such as $[CpMo(CO)(P_2N_2)]^+$ [26,27] and $[Mo(NO)(CO)(^iPr_2PCH_2CH_2)_2N)]^+$[24], for heterolytic activation of $H_2$. Although stoichiometric hydrogenation of olefins by Mo−H compounds using triflic acid as a proton source has been extensively studied[18,28–30], molybdenum complexes capable of catalyzing hydrogenation of unactivated olefins under mild conditions are scarce[31,32]. For example, Berke reported a highly efficient co-catalyst system $[(^iPr_2PCH_2CH_2)_2PPh)MoH(NO)(\eta^2-C_2H_4)]$(II)/Et$_3$SiH/ $B(C_6F_5)_3$, which achieved catalytic hydrogenation of olefins with maximum turnover frequencies (TOFs) up to 5250 h$^{-1}$ [33]. Beller found that low-valent molybdenum pincer complexes $[(^iPr_2PCH_2CH_2)_2NH) Mo(L)(CO)_2]$ (III), once activation by NaHBEt$_3$, serve as effective catalysts for the hydrogenation of styrenes[34]. Notably, Chirik reported a serial of pyridine(diimine) molybdenum complexes ($^{iPr}$PDI) $Mo(CH_2SiMe_3)_2$ (IV) and phosphino(imino)pyridine molybdenum cyclooctadiene complexes (PIP)Mo(COD) (V), have demonstrated exceptional performance in catalyzing hydrogenation of arenes and

[1]School of Chemistry and Chemical Engineering, Shandong University, 250100 Jinan, China. [2]College of Chemistry, Beijing Normal University, 100875 Beijing, China. [3]Davenport Chemical Research Laboratories, Department of Chemistry, University of Toronto, Toronto, ON M5S 3H6, Canada. ✉e-mail: d.song@utoronto.ca; wwg@bnu.edu.cn

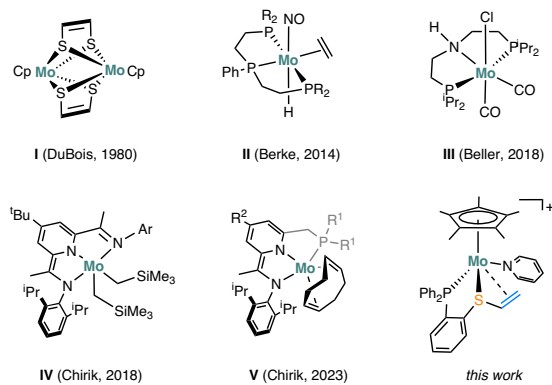

**Fig. 1 | Molybdenum-based olefin hydrogenation catalysts.** Selected literature examples I–V and the complex described in this work.

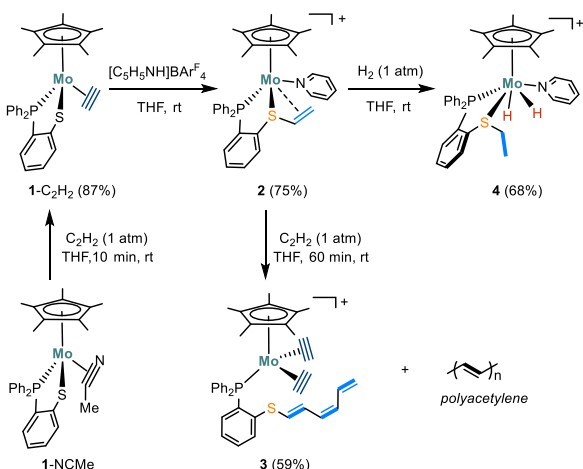

**Fig. 2 | Reactivity of mononuclear molybdenum-sulfur complexes.** Activation of $C_2H_2$ and $H_2$ by and interconversion among mononuclear molybdenum-sulfur complexes.

hindered olefins[35–37]. They discovered that the addition of $H_2$ to complex V results in [(PIP)MoH($\eta^5$-$C_6H_7$)], which is essential for the hydrogenation of benzene to cyclohexane[36].

Our group is interested in investigating the potential of metal-sulfur complexes, Cp*M(1,2-$Ph_2PC_6H_4S$), for bond activations and catalytic applications[38–40]. In the previous studies, we reported the high catalytic activity of [Cp*Mo(1,2-$Ph_2PC_6H_4S$)($\eta^2$-NCMe)] (**1**-NCMe) in the transfer hydrogenation of nitriles to primary amines[40]. In this study, we report the protonation of the acetylene complex, [Cp*Mo(1,2-$Ph_2PC_6H_4S$)($\eta^2$-$C_2H_2$)], (**1**-$C_2H_2$) to yield a cationic Mo(II) catalyst, [Cp*Mo(1,2-$Ph_2PC_6H_4S$ − CH = $CH_2$)(Py)]$^+$ (**2**), which is capable of catalyzing hydrogenation of various olefins at room temperature without the need for additives or external bases (Fig. 2). Upon protonation with pyridinium salts, the $C_2H_2$ ligand in **1**-$C_2H_2$ undergoes coupling with the thiolate ligand, resulting in the formation of **2** which contains a formed $\eta^3$-vinylthioether ligand. This cationic Mo(II) complex displays intriguing reactivity toward $C_2H_2$ and $H_2$ under ambient pressure, leading to diverse molybdenum species that have been isolated and crystallographically characterized. The reaction mechanisms were elucidated by combining experimental results with DFT calculations.

## Results

### Molybdenum acetylene complex

When a solution of **1**-NCMe in THF was exposed to $C_2H_2$ (1 atm), its color changed from brown to dark green within 10 min and the resulting molybdenum acetylene complex, **1**-$C_2H_2$ was isolated in ~87% yield (Fig. 2). The $^{31}$P NMR spectrum of **1**-$C_2H_2$ displayed a singlet at 97.5 ppm, slightly shifted compared to the signal at 103.0 ppm for **1**-NCMe. At room temperature, the $^1$H NMR spectrum of **1**-$C_2H_2$ exhibited two sets of broad resonances at 10.49 and 9.48 ppm, indicating the dynamic behavior of the $\eta^2$-$C_2H_2$ moiety binding at the Mo center[15]. Upon lowering the temperature to 0 °C, these resonances became well-resolved, and the signal at 9.48 ppm split into a doublet due to the coupling to the phosphorus atom ($J_{P-H}$ = 20 Hz)[41]. Analysis of the $^1$H–$^{13}$C HSQC spectrum revealed that the acetylenic protons at 10.49 and 9.48 ppm respectively correlate with the $^{13}$C signals at 190.3 and 184.9 ppm. Furthermore, the solid-state molecular structure of **1**-$C_2H_2$ was confirmed by X-ray crystallography (Fig. 3a). The acetylene ligand is bound to the Mo center in a side-on fashion and the Mo−C distances are 2.050(3) and 2.018(3) Å. The coordinated C ≡ C bond length (1.297(4) Å) is close to that reported for [Mo($t$-BuS)$_2$($t$-BuNC)$_2$(HC ≡ CH)] (1.28(2) Å)[13].

The protonation of **1**-$C_2H_2$ with one equiv. of the pyridinium salt [$C_5H_5$NH]BAr$^F_4$ in THF (BAr$^F_4$ = (3,5-$(CF_3)_2C_6H_3)_4$B, p$K_a$ = 5.5)[42] led to an immediate color change from green to brown. The $^{31}$P NMR spectrum suggests the complete conversion of **1**-$C_2H_2$ into a molybdenum species **2** ($m/z$ = 632.1438), which has a $^{31}$P NMR signal at 80.1 ppm.

Crystallographic analysis reveals that **2** is [Cp*Mo(1,2-$Ph_2PC_6H_4S$ − CH = $CH_2$)(Py)]BAr$^F_4$ (Fig. 3b). According to the crystal structure, the protonation of the acetylene ligand in **1**-$C_2H_2$ has led to the reductive coupling of the resulting vinyl ligand and the thiolate. The resultant vinylthioether unit, $\eta^3$-$CH_2$ = CHSAr, is attached at the metal center with unequal Mo−C distances of 2.241(3) and 2.181(3) Å. The metal-bound side-on olefin in **2** exhibits a C = C bond length of 1.419(4) Å, comparable to those reported for Mo(PNP)($C_2H_4$)$_3$ (1.416(2) and 1.425(2) Å)[43]. The $^1$H NMR spectrum of **2** in $d_8$-THF displays a distinct signal at 5.37 ppm (m, 1H) for the $\alpha$-C$H$ (with respect to the S atom) of the vinyl group, which gives an HSQC cross peak with the $^{13}$C signal at 83.9 ppm. The two protons on the $\beta$-carbon of the vinyl group are inequivalent and resonate at 1.68 and 1.09 ppm, respectively, and are correlated with the $^{13}$C signal observed at 32.5 ppm for the $\beta$-C in the HSQC spectrum.

### Activation of acetylene

When a solution of **2** was exposed to acetylene gas (1 atm) at room temperature, the brown solution gradually turned to yellow, with the concomitant formation of black precipitates. The resulting solid was identified to be polyacetylene (PA) by the IR spectrum[44]. It exhibits the characteristic $v_{C-H}$ signals at 2921, 1348, 1020 and 736 cm$^{-1}$ and $v_{C=C}$ signal at 1600 cm$^{-1}$ (Supplementary Fig. 16), which are indicative of both *cis* and *trans* polyacetylene[45]. Besides the production of PA, the reaction also produced a molybdenum complex **3**, which displayed a peak at 61.7 ppm in the $^{31}$P NMR spectrum. The $^1$H NMR spectrum of **3** features four signals at 10.66, 10.46, 8.99, and 8.73 ppm, suggesting two inequivalent side-on acetylene ligands.

Crystallographic analysis reveals the solid-state structure of **3**, which indeed contains two $\eta^2$-$C_2H_2$ ligands in the first coordination sphere of the Mo center (Fig. 3c). The C−C bond lengths of the two $\eta^2$-$C_2H_2$ ligands are 1.253(6) and 1.285(6) Å, which is consistent with the observation of two inequivalent side-on acetylene ligands in the NMR spectra. These C−C lengths are comparable to those found for the molybdenum acetylene complexes of Mo($C_2H_2$)(dppe)$_2$ (1.265(7) Å)[41], [MoO($C_2H_2$)(6-MePyS)$_2$] (1.2649(17) Å)[15], and Mo($C_2H_2$)($t$-BuS)$_2$($t$-BuNC)$_2$ (1.28(2) Å)[13]. The transformation of **2** to **3** involves not only the displacement of the pyridine ligand from the Mo center but also an unusual extension of the vinyl moiety by two $C_2H_2$ units and the dissociation of the resulting polyenylthioether motif from the Mo center. The elongation of the vinyl motif is reminiscent of the chain propagation reaction in polymerization reactions. Intriguingly, the bis(acetylene) complex **3** is also capable of polymerizing $C_2H_2$ albeit much less active than **2**. When a frozen solution of **3** (1 mM) in THF was treated

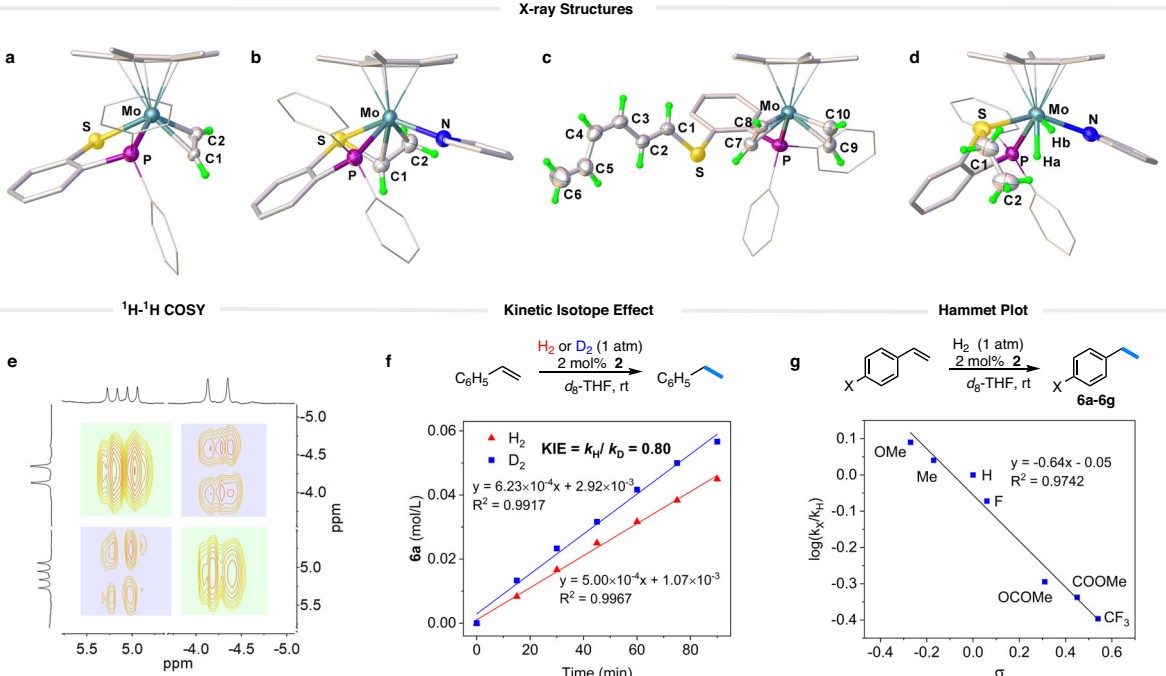

**Fig. 3 | Structural characterizations and hydrogenation of alkenes.** Crystal structures of **a** 1-$C_2H_2$, **b 2**, **c 3**, and **d 4** with the thermal ellipsoids shown at 50% probability level. For clarity, the portions of the molecules are shown in the wire-frame and stick style, while the BAr$^f_4$ anion and all hydrogen atoms on the Cp* and aromatic rings are omitted. Selected bond distances (Å) and angles (deg): for **1**-$C_2H_2$, Mo−S 2.3565(6); Mo−P 2.4523(6); Mo−C1 2.050(3); Mo−C2 2.018(3); C1−C2 1.297(4); H−C1−C2 145.0(3); H−C2−C1 143.6(3); for **2**, Mo−S 2.4183(7); Mo−P 2.4904(7); Mo−N 2.274(2); Mo−C1 2.181(3); Mo−C2 2.241(3); C1−C2 1.419(4); for **3**, Mo−P 2.5490(9); Mo−C7 2.097(4); Mo−C8 2.044(4); Mo−C9 2.122(4); Mo−C10 2.065(4); C7−C8 1.253(6); C9−C10 1.285(6); for **4**, Mo−S, 2.3977(16); Mo−P 2.459(3); C1−C2 1.520(12). **e** Structural characterization of **4** by $^1$H−$^1$H COSY. **f** Time-course of ethylbenzene production in styrene hydrogenation with $H_2$ and $D_2$ catalyzed by **2** (2 mol%). **g** Hammet plot for the hydrogenation of *para*-substituted styrenes.

with $C_2H_2$ gas (1 atm) and then warmed up to 40 °C, a black precipitate of PA gradually formed over 24 h.

## Activation of H$_2$

Although oxidative addition of $H_2$ to cationic molybdenum complexes was proposed as a fundamental step in catalytic hydrogenation of C = O bonds[20,22,25,30], the resulting cationic dihydride species has not been directly observed for the reaction of cationic Mo(II) complexes with $H_2$[18,20,23,46,47]. Regarding the Mo(IV) dihydride compounds, Nikonov reported [(ArN)Mo(H)$_2$(PMe$_3$)$_3$] arising from the reaction of [(ArN)Mo(H)(Cl)(PMe$_3$)$_3$] with L-selectride, which is active for catalytic hydroboration[48,49]. Surprisingly, complex **2** reacts with $H_2$ to afford an active Mo(IV) dihydride for hydrogenation of unactivated C = C bond.

The exposure of **2** (in $d_8$-THF) to $H_2$ (1 atm) resulted in the formation of a molybdenum dihydride compound **4**, which exhibits a sharp $^{31}$P resonance at 76.1 ppm[27,50]. The production of **4** was initially confirmed by high-resolution mass spectroscopy (HRMS), where a strong ionic peak at $m/z$ = 636.1735 was observed (Supplementary Fig. 14), in comparison to that of 632.1438 found for **2**. When $D_2$ was employed for the reaction, HRMS analysis of the reaction mixture revealed an ionic peak at 640.2057 (Supplementary Fig. 15). This finding can be rationalized by the addition of two molecules of $D_2$ to compound **2** through hydrodeuteration of the vinyl moiety and oxidative addition of $D_2$ to the molybdenum center. At room temperature, the hydride resonances of **4** coalesced into the baseline in the $^1$H NMR spectrum[51]. However, upon cooling the $C_6D_5Cl$ solution to 253 K, the hydride resonances appear as well-resolved peaks in the $^1$H NMR spectrum. One hydride ligand shows a characteristic upfield signal at −4.19 ppm as a doublet, while the signal of the other hydride is much further downfield with a chemical shift of 5.16 ppm (dd)[52]. The $^1$H−$^1$H COSY spectrum recorded at 253 K shows that the two sets of hydride

signals correlate with each other with an large $^2J_{H-H}$ of 110 Hz (Fig. 3e)[51]. Based on $^1$H and $^{31}$P NMR analysis, the hydride signal at low field is coupled to the phosphorus nuclei with $^2J_{P-H}$ = 55 Hz, whereas no phosphorus-hydrogen coupling was observed for the hydride at the higher field. Upon hydrogenation, the olefin ligand in **2** is converted into a non-coordinating ethyl group as indicated by the $^1$H NMR signals at 3.42 ppm (m, 1H), 2.56 ppm (m, 1H) and 1.17 ppm (m, 3H). It is important to mention that complex **4** is unstable in solution and undergoes gradual degeneration, resulting in the formation of unidentified species.

Single-crystal X-ray diffraction confirms the solid-state structure of **4** (Fig. 3d), which is a cationic Cp*Mo(IV) dihydride species with a phosphino-thioether and a pyridine ligand. In general, the coordination geometry of **4** is similar to those eight-coordinate molybdenum analogs such as [CpMo(CO)(P$^{Et}$N$^{Me}$P$^{Et}$)(H)$_2$]$^+$ [27], [Cp*MoH$_2$(dppe)(MeCN)]$^+$ [51], and [CpMoH$_2$(PMe$_3$)$_3$]$^+$ [50]. The two hydride ligands were located from the difference Fourier map and refined isotropically. The Mo−H bond lengths were found to differ by ca. 0.2 Å, with Mo−H$_a$ = 1.63(8) vs. Mo−H$_b$ = 1.84(5) Å[53]. The H$_a$−Mo−P and H$_b$−Mo−P angles are 66(3)° and 123.8(17)°, respectively. These bond and angle parameters are indicative of two inequivalent hydrides binding at the Mo center. The DFT optimized structure of compound **4**, featuring two hydrides, closely resembles the X-ray structure. Computationally determined chemical shifts for these hydrides are −3.7 and 2.9 ppm, respectively, which align with the distinct chemical shifts observed experimentally. According to the computations, the hydride closer to the P-donor is more deshielded, resulting in a downfield signal. Additionally, the hydrogenation of **2** to **4** resulted in a noticeable elongation of the C1−C2 bond from 1.419(4) Å to 1.520(12) Å, indicating the reduction of the vinyl group to an ethyl group[47]. In contrast, the Mo−P and Mo−S distances in **4** change only slightly upon hydrogenation compared to those in **2**.

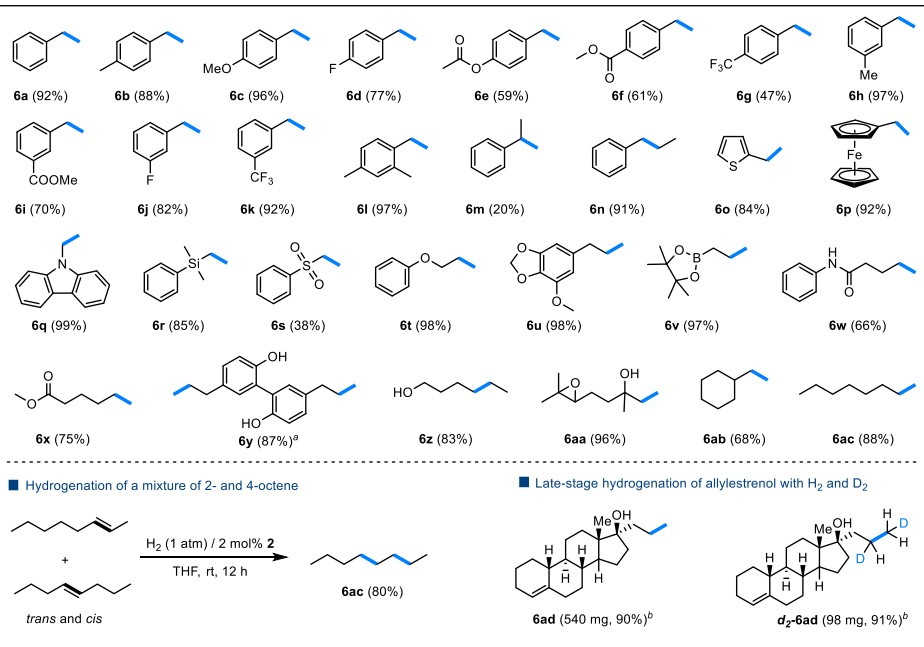

**Fig. 4 | Substrate scope.** Reaction conditions: alkene substrate (**5**) (0.2 mmol), catalyst (**2**) (2 mol%), 1,3,5-trimethoxybenzene (0.067 mmol) as internal standard, $H_2$ (1 atm) in 0.6 mL $d_8$-THF, rt, 12 h. Yields were determined by $^1H$ NMR spectroscopy. $^a$**2** (4 mol%). $^b$Isolated yields.

## Catalytic hydrogenation of alkenes

The facile reaction of **2** with $H_2$ to form a dihydride prompted us to examine the catalytic performance of **2** toward styrene hydrogenation. In a J. Young NMR tube, a solution of styrene (0.2 mmol), and **2** (4 μmol, 2 mol%) in $d_8$-THF was degassed and exposed to $H_2$ (1 atm) at room temperature. The reaction gives ethylbenzene (**6a**) in 92% yield (with TON ≈ 46) in 12 h based on the integration of NMR signals relative to that of the internal standard. The Mo-catalyzed addition of $H_2$ to the C = C bond was also confirmed by a deuterium labeling experiment. When $D_2$ was used for the hydrogenation of 4-methylstyrene, the reaction unambiguously gave the corresponding deuterated product PhCHDCH$_2$D ($d_2$-**6a**) (Supplementary Fig. 3). Figure 3f depicts the time-dependent production of ethylbenzene during the initial stage of styrene hydrogenation, utilizing $H_2$ and $D_2$ as reactants, respectively. The hydrogenation rate with $D_2$ (1 atm, $k_D = 6.23 \times 10^{-4}$ M min$^{-1}$) is faster than that with $H_2$ (1 atm, $k_H = 5.00 \times 10^{-4}$ M min$^{-1}$) at 25 °C, resulting in an inverse kinetic isotope effect (KIE) of 0.80. This implies that the turnover-determining transition state involves the strengthening of a C − H bond[54].

The substrate scope of the hydrogenation reaction was then investigated at a 2 mol% loading of **2** (Fig. 4). The catalytic hydrogenation of styrenes with Me and OMe groups at the *para*-position gave **6b** and **6c** in 88 and 96% NMR yields, respectively. In contrast, *para*-F, *para*-OCOMe, *para*-COOMe, and *para*-CF$_3$ groups lower the yield from 77% to 47% (**6d**–**6g**). To evaluate the influence of electronic variation in the substituent on the reaction rate, kinetic studies were performed on hydrogenation of the *para*-substituted styrene derivatives (*p*-X-styrene, **5a**–**5g**). The reaction progress was monitored using $^1H$ NMR spectroscopy, revealing a linear reaction profile within the initial 4 h (Supplementary Fig. 10). The reaction rate was found to strongly depend on the electronic nature of the *para*-substituent. The kinetic data are correlated with the standard Hammett σ$_{para}$ values[55], resulting in a negative slope of ρ = −0.64 (Fig. 3g). The small absolute value of the Hammett electronic parameter suggests that the reaction site in the turnover-limiting transition state is more remote than the benzylic carbon from the *para*-X-group, ruling out the insertion step being the turnover-limiting[56]. The small negative ρ is consistent with the reductive elimination step being

turnover-limiting, where the terminal carbon of the styrene substrate is involved.

Although no clear trend was observed for styrenes bearing various *meta*-substituents, the hydrogenation reactions of *meta*-substituted styrenes give good to excellent yields (**6h**–**6k**). The introduction of steric bulk at the *ortho*-position of styrene has little impact on the reaction, as evidenced by the nearly quantitative reduction of 2,4-dimethylstyrene (**6l**). This molybdenum-catalyzed reaction is inhibited by α-substituent on the styrene, e.g., the hydrogenation of α-methyl-styrene gives cumene (**6j**) in only 20% yield, presumably because sterically encumbered 1,1-disubstituted olefins are poor ligands. In contrast, β-methylstyrene can be hydrogenated to give **6n** in a 91% yield. Not only styrenes, but also heteroaryl alkenes like 2-vinylthiophene and vinylferrocene can be also efficiently hydrogenated (**6o**, **6p**).

This Mo-based hydrogenation system demonstrated compatibility with activated alkenes bearing both carbazolyl (**6q**) and silyl groups (**6r**). Interestingly, the presence of a tethered sulfonyl group (**6s**) suppresses the hydrogenation of these alkenes. However, olefins bearing various oxygen-containing functional groups, such as ether (**6t**, **6u**), ester (**6x**), amide (**6w**), and alcohol (**6y**, **6z**) all underwent smooth hydrogenation. This is noteworthy as such substrates were rarely tolerated by Mo-based catalysts[57,58]. Also, the boronic ester functionality was well-tolerated in the hydrogenation of allylic derivative (**6v**). The catalysis demonstrates high selectivity and good tolerance towards functional groups, as exemplified by the efficient hydrogenation of 6,7-epoxy-linalool to **6aa** with both the epoxy and hydroxyl groups remaining intact. Unactivated olefins, such as vinyl-cyclohexane and 1-octene, can be hydrogenated into the corresponding alkanes **6ab** and **6ac** in 68% and 88% yield, respectively. Notably, the hydrogenation of a mixture of 2-octene and 4-octene gives **6ac** (80% yield) exclusively, i.e., no isomerization products.

The mild Mo-catalyzed protocol also offers practical applications for the late-stage hydrogenation and hydrodeuteration of complex bioactive molecules. For instance, when allylestrenol (**5ad**), a progestin medication, was subjected to hydrogenation, the terminal unactivated C = C double bond was selectively reduced[59]. The reaction carried out on a 2 mmol scale under catalytic conditions, afforded the desired

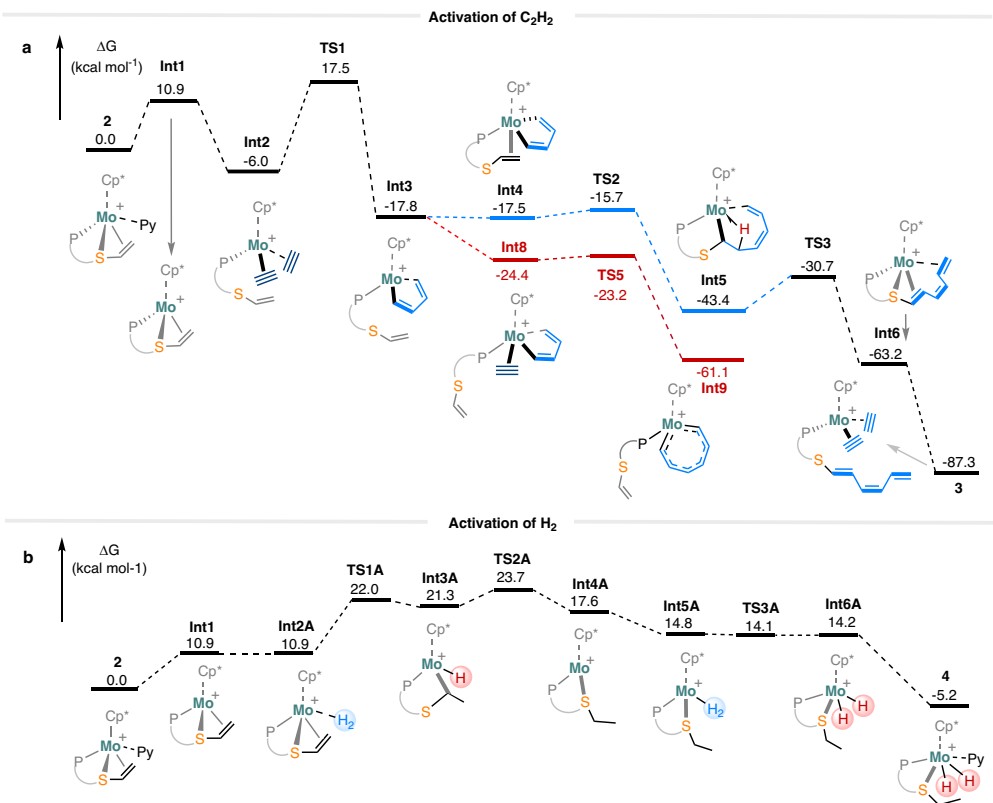

**Fig. 5 | Gibbs free energy diagram (in kcal mol⁻¹, at 298 K, 1 M concentration, and 1 atmosphere pressure) for the reaction of 2 with a C₂H₂ and b H₂.** For computational details, see Supplementary Information.

product (**6ad**) with an impressive 90% isolated yield. Furthermore, by utilizing D₂, deuterium was successfully incorporated through hydrodeuteration of the allyl group. As a result, the reaction efficiently delivered the deuterated product, **d₂‑6ad**, featuring a D-labeled ethyl group, without significant loss in yield (98 mg, 91% yield).

## Discussion

As a cationic Mo(II) complex, **2** exhibits intriguing reactivity toward C₂H₂ and H₂ under ambient pressure. To elucidate the reaction mechanisms, DFT calculations were conducted to investigate the transformations of **2** to **3** and **4**, respectively. As shown in Fig. 5a, the reaction of **2** with C₂H₂ could begin with the dissociation of a pyridine ligand from **2** to afford **Int1** (10.9 kcal mol⁻¹), which binds with two acetylene molecules to form the bis(acetylene) intermediate (**Int2**) (−6.0 kcal mol⁻¹). The two acetylene ligands undergo cycloaddition to form a five-membered metallacycle, (**Int3**) (−17.8 kcal mol⁻¹), via the rate-determining transition state (**TS1**) (17.5 kcal mol⁻¹). Starting from **Int3**, the coordination of the alkene sidechain of the phosphine ligand leads to the formation of **Int4** (−17.5 kcal mol⁻¹) on the pathway to **3**, while the competitive coordination of acetylene gives **Int8** (−24.4 kcal mol⁻¹) toward chain growth of polyacetylene.

Proceeding toward the formation of **3, Int4** undergoes alkene insertion to form **Int5** (−43.4 kcal mol⁻¹) via transition state **TS2** (−15.7 kcal mol⁻¹) (blue in Fig. 5a)[60]. **Int5** features a 7-membered metallacycle with β-agostic interaction, which sets up the β-C-H bond for the subsequent σ-bond metathesis to give **Int6** (−63.2 kcal mol⁻¹) via transition state **TS3** (−30.7 kcal mol⁻¹). Finally, two acetylene ligands displace the triene sidechain from the metal center of **Int6** to give **3** (−87.3 kcal mol⁻¹). It is conceivable that the two acetylene ligands in **3** could also undergo cycloaddition to form a five-membered metallacycle intermediate (**Int7**) (−93.5 kcal mol⁻¹), which can react with acetylene, leading to the formation of polyacetylene. However, the cycloaddition has a free energy barrier of 27.5 kcal mol⁻¹, consistent

with the observation that the isolated sample of **3** gives a much slower acetylene polymerization, even at 40 °C when compared to **2**. In the reaction mixture of **2** and acetylene, however, since the C–C bond formation reactions are very exothermic, the temperature of the mixture might be sufficiently high for **3** to polymerize acetylene.

The formation of **Int8** from **Int3** is exergonic by 6.6 kcal mol⁻¹, whereas the formation of **Int4** is slightly endergonic by 0.3 kcal mol⁻¹. Consequently, the formation of **Int8** is favored (red in Fig. 5a). Subsequently, **Int8** can undergo facile (i.e., 0.8 kcal mol⁻¹ free energy barrier) insertion of acetylene to give **Int9** (−61.1 kcal mol⁻¹) via transition state **TS5** (−23.2 kcal mol⁻¹). It is conceivable that **Int9** can keep inserting more acetylene in a similar facile fashion to form PA. Clearly, the polymer chain propagation pathway (in red) is favored both thermodynamically and kinetically, whereas the formation of **3** is slower (i.e., with a barrier of 12.7 kcal mol⁻¹ from **Int5** to **TS3**). Based on our calculations, the formation of **Int3** is the slow step. When a small fraction of **Int2** turns into **Int3**, the fast polymerization initiated by **Int3** depletes C₂H₂ quickly. The remainder of **Int2** will just turn into **3** after the C₂H₂ concentration drops below a certain threshold, which is consistent with the relatively high isolated yield for **3** from the polymerization experiment.

For the transformation of **2** to **4**, the mechanism was also investigated by DFT calculations. As shown in Fig. 5b, pyridine dissociation from **2** gives **Int1** (10.9 kcal mol⁻¹), which undergoes thermally neutral H₂ coordination to form **Int2A** (10.9 kcal mol⁻¹). The subsequent dihydrogen splitting between the terminal carbon of the olefin ligand and the metal center via **TS1A** (22.0 kcal mol⁻¹) gives the hydridoalkyl Mo(IV) intermediate **Int3A** (21.3 kcal mol⁻¹), which can undergo facile reductive elimination via **TS2A** (23.7 kcal mol⁻¹) to form **Int4A** (17.6 kcal mol⁻¹). No stable dihydride intermediate was located between **Int2A** and **Int3A**. In **TS1A** the distance between the two hydrogen atoms originated from H₂ is 1.51 Å, shorter than that of a typical dihydride; **TS1A** is most consistent with the heterolytic splitting

of a coordinated $H_2$ between the olefin carbon and the metal center, i.e., deprotonation of the $H_2$ ligand by the alkene ligand. The coordination of $H_2$ onto the Mo center of **Int4A** gives **Int5A** (14.8 kcal mol$^{-1}$), which undergoes oxidative addition via **TS3A** (14.1 kcal mol$^{-1}$) to give the dihydride intermediate **Int6A** (14.2 kcal mol$^{-1}$). The coordination of Py converts **Int6A** to **4** (−5.2 kcal mol$^{-1}$). All intermediates are unstable along this reaction pathway, which is consistent with the fact that no intermediate was observed experimentally. Additionally, experimental studies have revealed that an excess of pyridine impedes the conversion of **2** to **4** and the catalytic hydrogenation of styrene (Supplementary Figs. 5 and 6). These findings correspond with the DFT results, indicating the importance of pyridine dissociation in the hydrogenation processes. In particular, when employing [Cp*Mo(1,2-Ph$_2$PC$_6$H$_4$S − CH = CH$_2$)(2,6-lutidine)]BAr$^F_4$, which is in-situ generated by **1**-C$_2$H$_2$ with 2,6-lutidinium salts, as the catalyst, the hydrogenation of styrene with $H_2$ at room temperature for 3 h results in a slightly higher yield of **6a** (35%) in comparison to the 26% yield obtained in the styrene reduction catalyzed by **2** under identical reaction conditions (Supplementary Figs. 7 and 9).

Given that our DFT calculations show the involvement of a few reactive intermediates, **Int4A−Int6A,** in the reaction of **2** and $H_2$, we envision that the catalytic hydrogenation reaction likely shares these intermediates in the same reaction sequence. The proposed catalytic mechanism is shown in Fig. 6. The alkene substrate coordinates to the metal center of **Int6A** to afford **Int7A**. The subsequent alkene insertion into a Mo−H bond forms a hydrido alkyl species, **Int8A**, which then

releases the alkane product via reductive elimination and regenerates **Int4A**. Alternatively, the alkene substrate may coordinate with the metal center of **Int4A** to form **Int9A**, which then activates $H_2$ to form **Int7A** via a dihydrogen complex **Int10A**.

The Gibbs free energy profiles of these reaction pathways are shown in Fig. 7. Starting from **Int4A**, the styrene coordination pathway (red in Fig. 7) is more energetically favorable than the $H_2$ activation pathway (blue in Fig. 7). Both pathways are facile at ambient temperature and lead to the same alkene dihydride complex **Int7A**. The energetic span of the catalytic cycle computed under the standard conditions is 20.8 kcal mol$^{-1}$ with the turnover-determining states being **Int7A** and **TS5A** (the transition state of the reductive elimination step for product release). This result is consistent with the small negative $\rho$ value from the Hammett plot. The computed energetic span between $d_2$-**Int7A** and $d_2$-**TS5A** was 20.6 kcal mol$^{-1}$, which is consistent with the experimentally measured inverse KIE. Based on such an energetic span, one would expect much faster catalytic hydrogenation at ambient temperature. However, all the unsaturated catalytic intermediates are in equilibrium with the more stable pyridine-bound off-cycle species. For example, **Int4A**, **Int5A**, and **Int6A** are 17.4, 18.2, and 19.4 kcal mol$^{-1}$ less stable than the corresponding pyridine-bound species, respectively, i.e., the concentrations of the active species are very low (Supplementary Fig. 62). This hypothesis is supported by our experimental observation that additional pyridine retards the catalytic hydrogenation (Supplementary Fig. 6).

In summary, we have demonstrated the protonation of a molybdenum(II)-acetylene complex using a pyridinium salt, resulting in the formation of a multifunctional cationic Mo(II)-vinylthioether complex. This complex exhibits efficient catalytic activity for hydrogenation of various aromatic and aliphatic alkenes at ambient temperature. The reaction exhibits excellent substrate compatibility and high functional group tolerance without the need for any additives. Mechanism studies reveal that the catalysis involves a Mo(IV) dihydride intermediate arising from an uncommon oxidative addition of $H_2$ to the cationic Mo(II) center. Additionally, we found that complex **2** displayed intriguing reactivity toward C$_2$H$_2$, causing the vinylthioether ligand to undergo chain propagation and yield a dangling triene chain. Our future studies will focus on modifying the coordination sphere of [Cp*Mo(1,2-Ph$_2$PC$_6$H$_4$SR)] to enhance the catalytic performance in the transformation of C$_2$H$_2$ and substituted alkynes.

## Methods
### General procedure for catalytic hydrogenation of alkenes
A J. Young NMR tube charged with **2** (6 mg, 0.004 mmol), alkenes (0.2 mmol) in $d_8$-THF (0.6 mL), was added 1,3,5-trimethoxybenzene (11.2 mg, 0.067 mmol) as the internal standard. The tube was taken out from the glovebox and immersed in a liquid nitrogen bath, and gently

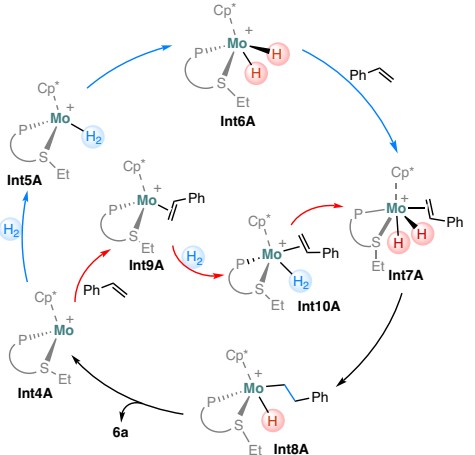

**Fig. 6 | Proposed mechanism for the Mo-catalyzed alkene hydrogenation.** Blue path: $H_2$ activation precedes olefin coordination. Red path: olefin coordination precedes $H_2$ activation.

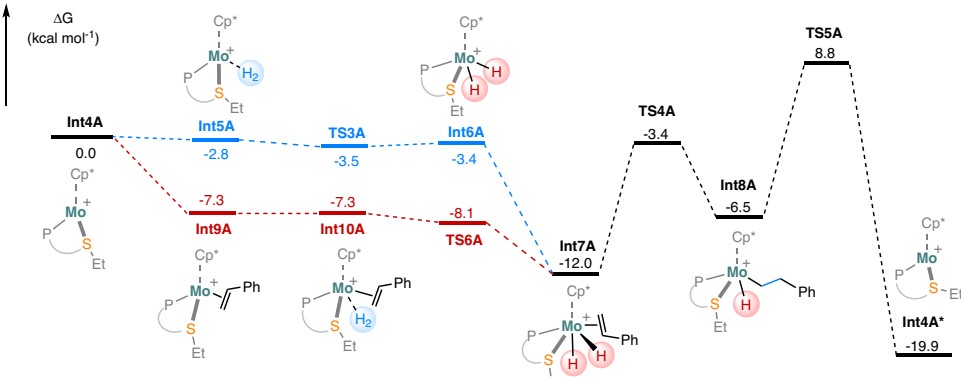

**Fig. 7 | Gibbs free energy diagram (in kcal mol$^{-1}$, at 298 K, 1 M concentration, and 1 atmosphere pressure) for the Mo-catalyzed hydrogenation of styrene.** Blue path: $H_2$ activation precedes olefin coordination. Red path: olefin coordination precedes $H_2$ activation. For computational details, see Supplementary Information.

degassed under vacuum. The solution was then warmed up to room temperature and pressurized with $H_2$ gas (1 atm). After reaction at room temperature for 12 h, the sample was analyzed by $^1$H NMR to determine the yield of the hydrogenated product.

## Data availability

Crystallographic data for the structures reported in this Article have been deposited at the Cambridge Crystallographic Data Centre, under deposition numbers CCDC 2245911 (1·$C_2H_2$), 2245912 (2), 2245914 (3) and 2245913 (4). Copies of the data can be obtained free of charge via https://www.ccdc.cam.ac.uk/structures/. All other data related to experimental procedures, compound characterization, mechanistic experiments, and spectra are available within the paper and Supplementary Information, or from the corresponding author upon request. Coordinates of the optimized structures are provided in the source data file. Source data are provided with this paper.

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

## Acknowledgements

We thank the financial support from National Natural Science Foundation of China (22022102 and 22071010) and Natural Science Foundation of Shandong Province (ZR2019ZD45). D.S. thanks the Natural Sciences and Engineering Research Council (NSERC) of Canada for funding. This research was enabled in part by support provided by Sharcnet (www.sharcnet.ca) and Compute Canada (www.computecanada.ca).

## Author contributions

W.W., D.S., M.X., and C.-H.T. conceived and designed the project and wrote the manuscript. M.X. conducted the experiments and analyzed the data. Z.P. and K.T. carried out the IR and mass spectrometry measurements. M.X. performed the characterization with support from J.J. using Low-Temperature NMR Spectroscopy. D.S. carried out the computational analyses. All authors provided comments on the experiments and the manuscript during its preparation.

## Competing interests

The authors declare no competing interests.
