## [Peer Review File · Nature Communications]

Catalytic Hydrogenation of Olefins by a Multifunctional Molybdenum-Sulfur ComplexReviewers' Comments:

Reviewer #1:

Remarks to the Author:

The manuscript by Song, Wang and coworkers describes a molybdenum complex for catalytic hydrogenation of olefins. They found a molybdenum(II) Cp* complex which is coordinated by a bidentate sulfido phosphine ligand and acetylene (2). Protonation of the latter leads finally to a coordinated vinyl thioether. Reactivity towards H₂ or C₂H₂ led to unusual complexes, namely a dihydrido complex and a thioether complex with a C₆H₇ chain at sulfur derived from two additional molecules of acetylene. Compound 2 proved to be a very good catalyst for olefin hydrogenation at mild conditions. All compounds are convincingly characterized by spectroscopic and X-ray crystallographic means and the mechanisms of compound formation supported by DFT calculations. The clearness of the manuscript is exceptionally high.

It was a pleasure to read the manuscript as it describes beautiful organometallic chemistry. Furthermore, the found catalytic activity under impressively mild conditions is of high interest. Molybdenum is quite earth abundant and thus economically interesting and it is biologically relevant (only metal of the second row of the transition metals) and therefore virtually non-toxic.

For these reasons I consider the research highly significant for researchers interesting in catalysis and the manuscript highly suitable for the high-class journal Nature Communication so that I recommend publication.

However, prior to that there are some issues/comments that need to be addressed:

- i) I am aware that high-resolution MS is considered a mean to support the purity of the bulk material, however I am a fan of elemental analysis. This is particularly true in case of molybdenum where molybdates tend to be quite soluble. They would not be detectable by NMR spectroscopy nor would they influence the molecular ion in MS. Thus, the authors should additionally obtain elemental analysis data for reassurance.
- ii) Related to the comment above, all NMR data provided in the supporting information contain minor signals that are not assigned. Particularly in 4 but also in the others. It seems that NMR data of 4 is of a material prior to isolation. Why is that? I assume that stability is an issue. Thus, macroscopic behavior of 4, and best of all Mo compounds, must be described in the manuscript and the authors have to comment on the minor signals in all spectra.
- iii) Why is NMR data of 1 measured at 273 K and that of 4 at 253 K and not rt? This should be mentioned in the manuscript.
- iv) The chemical shift difference of the two hydrido ligands in 4 is extremely high which makes one wonder whether the low-field signal is indeed a hydride. Did the authors consider other options such as an η^1 -bound H₂? At the very least, a possible explanation for the large difference should be discussed.
- v) X-ray data of 3 and 4 must be discussed by referring to literature data not only to the respective compounds in the manuscript.
- vi) catalysis was performed with 2 only. I assume that 4 was not investigated for stability reasons. A comment must be provided.
- vii) The comment on the tolerance of Mo catalyst on page 9 I cannot support. The cited literature refers to Mo(VI) from e.g. Schrock which are of course oxophilic while compound 2 in oxidation state +II is far less oxophilic. Thus, please rephrase.
- viii) Figure 6: is the structure of Int9 possibly wrong? I cannot see how a molybdenum carbene can be formed? It is the product of acetylene insertion which gives a 7-membered metallacycle with three C-C double bonds but no metal-carbon double bond. Please comment on it.
- ix) Figure 6: it interesting that compound 3 is isolated in 59 % yield even though the barriers for polyacetylene formation are lower. Is 3 formed immediately or only after most acetylene has been converted to polyacetylene? Did the authors try to use stoichiometric amounts of acetylene?
- x) Figure 7: I find it surprising that between Int2A and Int3A no dihydride is found while later the

dihydride is isolated (even 68 % yield). If no dihydride is formed, how is the alkyl formed? Do the authors consider heterolytic splitting of H₂ forming H⁻ and H⁺ which protonates the vinyl? While this would be similar to reactions with acetylene, I doubt it. A dihydride seems the most likely TS. A comment on what TS1A might be, must be provided.

xi) Page 2: scare must be scarce.

xii) Since dissociation of pyridine the authors should consider using methyl pyridines or lutidines which could possibly speed up catalysis even more.

Reviewer #2:

Remarks to the Author:

The manuscript by Minghui Xue and coworkers reports a molybdenum-sulfur complex 2 that catalyzes acetylene polymerization (with substantial catalyst deactivation to 3, and potentially to other species) and catalyzes alkene hydrogenation through a Mo(IV) dihydride 4. The manuscript extends recent work by the Beller and Chirik groups and others, all of which was published in specialized chemistry journals (refs 31-37). DFT-predicted mechanisms for reaction of 2 with H₂ and with ethylene include rather high barriers to catalyst activation (Int2 to Int3, barrier 23.5 kcal/mol, TS3A, barrier 23.7 kcal/mol). The authors also include a speculative mechanism for alkene hydrogenation based on the DFT-predicted reaction of 2 with H₂. A detailed DFT mechanism would be a better methodology. It is not clear, from the authors' writeup, that this Mo-based alkene hydrogenation catalyst will outperform existing non-precious-metal alkene hydrogenation catalysts (see the 2015 review by Chirik). In my judgement, the manuscript has a modest significance as a follow-up to refs 31-37. It would be more appropriate as a submission to the specialized chemistry journals that published refs 31-37.

Reviewer #3:

Remarks to the Author:

The submitted work by Wenguang Wang and coworkers is not eligible for publication in Nature Communications. The paper deals with the hydrogenation of olefins facilitated by a Mo complex. The reduction of alkenes with gaseous hydrogen is a well-established catalytic transformation and Mo is well-known for its ability to catalyze redox processes. Furthermore, the substrate scope is rather narrow and the authors merely report NMR yields for their products. Hence, the given manuscript lacks the novelty which is necessary for publication in Nat. Commun.

However, the experiments including the calculations and the NMR studies were all well-performed and thus I recommend submission of the subject manuscript to a more catalysis-dedicated journal such as ChemCatChem.

Reviewer #4:

Remarks to the Author:

This manuscript reported a molybdenum complex for catalytic hydrogenation of various aromatic and non-activated aliphatic alkenes. As author reported, after protonated, 1-C₂H₂ converted to complex 2, which is the pre-catalyst for various alkenes hydrogenations. This complex can react with H₂ and form Mo(IV)-dihydride 4 which is related to the proposed mechanism. Various characteristic methods as well as DFT calculations were used for the compounds' characterizations, reactions analysis and mechanism study. This is a great topic and complex 2 is an intriguing hydrogenation catalyst; however, to be suitable for publication in Nature Communications, the manuscript needs to be improved, in particular, by addressing the following key aspects:

(1) The quality of characterization of Mo(IV)-dihydride 4 needs to be improved. The ¹H NMR spectrum looks not pure. For SXR, with alert A errors, and the R factor of 0.09, the H cannot be located

accurately. Please provide better crystallography data or other evidence. Authors may also try deuterium experiment to locate the hydrides. Besides, the authors assigns the two hydrides, one in upfield and the other in downfield, further references and characterizations need to be provided. Moreover, for all four molybdenum complexes, please identify the solvent residue signal, water peak, and impurities in $^1\text{H-NMR}$. For $^{31}\text{P-NMR}$, what is the internal or external standard? For ESI-MS, please also add the whole scope of the spectra first, and then give the zoom in of founded peaks.

(2) In catalytic hydrogenation section, the authors may check the Hammett σ parameters of the substituents, the yields percentage may be related to the electron density. The effects (such as electronic/steric) on alkenes' yields need to be further analyzed to associate the mechanism. The hydrogenation reaction were only studied by small scale in NMR tube. Did the authors try bulk catalysis and isolate the products? Bulk catalysis experiment can further test the catalytic ability of the Mo complex.

(3) In the hydrogenation mechanism cycle (Fig 8), besides Mo(IV)-dihydride, is there any other intermediates observed or confirmed by any wet experiments?

(4) The structure of the main-text needs to be improved. In discussion part, there are too much DFT details. Conclusions of all important experiments and future perspective need be re-written in better way.

Minor Questions/Suggestions:

(1) Please add percentage yields in Fig.2., and add the temperature in the step from 1-C₂H₂ to 2.

(2) Merge Figure 3 and 4 together or move figure 4 into Supplementary Information.

Response to Reviews: NCOMMS-23-32745

Reviewer #1 (Remarks to the Author):

The manuscript by Song, Wang and coworkers describes a molybdenum complex for catalytic hydrogenation of olefins. They found a molybdenum(II) Cp* complex which is coordinated by a bidentate sulfido phosphine ligand and acetylene (2). Protonation of the latter leads finally to a coordinated vinyl thioether. Reactivity towards H₂ or C₂H₂ led to unusual complexes, namely a dihydrido complex and a thioether complex with a C₆H₇ chain at sulfur derived from two additional molecules of acetylene. Compound 2 proved to be a very good catalyst for olefin hydrogenation at mild conditions. All compounds are convincingly characterized by spectroscopic and X-ray crystallographic means and the mechanisms of compound formation supported by DFT calculations. The clearness of the manuscript is exceptionally high.

It was a pleasure to read the manuscript as it describes beautiful organometallic chemistry. Furthermore, the found catalytic activity under impressively mild conditions is of high interest. Molybdenum is quite earth abundant and thus economically interesting and it is biologically relevant (only metal of the second row of the transition metals) and therefore virtually non-toxic.

For these reasons I consider the research highly significant for researchers interesting in catalysis and the manuscript highly suitable for the high-class journal Nature Communication so that I recommend publication.

Response: We thank this reviewer for the positive comments and for supporting the publication of our work in Nature Communication.

However, prior to that there are some issues/comments that need to be addressed:

i) I am aware that high-resolution MS is considered a mean to support the purity of the bulk material, however I am a fan of elemental analysis. This is particularly true in case of molybdenum where molybdates tend to be quite soluble. They would not be detectable by NMR spectroscopy nor would they influence the molecular ion in MS. Thus, the authors should additionally obtain elemental analysis data for reassurance.

Response: We appreciate the suggestion from this reviewer. We have carried out elemental analysis for complexes 1-C₂H₂, 2, 3, and 4 during the revision process. We have included the resulting data in the "Synthesis and Characterization" section for each corresponding complex within the Supplementary Information. For more detailed information, please refer to Supplementary Information pages S4-S6.

ii) Related to the comment above, all NMR data provided in the supplementary information contain minor signals that are not assigned. Particularly in **4** but also in the others. It seems that NMR data of **4** is of a material prior to isolation. Why is that? I assume that stability is an issue. Thus, macroscopic behavior of **4**, and best of all Mo compounds, must be described in the manuscript and the authors have to comment on the minor signals in all spectra.

Response: We sincerely appreciate the valuable feedback provided by this reviewer. Some minor signals in ^1H NMR spectra have been appropriately assigned in Supplementary Information. As the reviewer pointed out, compound **4** was indeed unstable in solution. We have recollected NMR spectra of **4** in $\text{C}_6\text{D}_5\text{Cl}$ at 253 K. (Supplementary Fig. 28)

iii) Why is NMR data of **1** measured at 273 K and that of **4** at 253 K and not rt? This should be mentioned in the manuscript.

Response: Thank you for the queries. Please see the explanations in the main text.

For **1**, “At room temperature, the ^1H NMR spectrum of **1**- C_2H_2 displayed two sets of broad resonances for the acetylenic protons at δ 10.49 and 9.48, indicating the dynamic behavior of the $\eta^2\text{-C}_2\text{H}_2$ moiety binding at the Mo center.¹⁵ At 0 °C, however, these signals became well-resolved, and the signal at δ 9.48 split into a doublet due to the coupling to the phosphorus atom ($J_{\text{P-H}} = 20$ Hz).⁴¹ Further analysis of the ^1H - ^{13}C HSQC spectrum revealed that the acetylenic protons at δ 10.49 and δ 9.48 correlate with the ^{13}C signals at δ 190.3 and δ 184.9, respectively.”

Regarding the spectroscopic properties of **4**, we revised the statements as following:

“The exposure of **2** (in d_8 -THF) to H_2 (1 atm) resulted in the formation of a molybdenum dihydride compound **4**, which exhibits a sharp ^{31}P resonance at δ 75.9.^{27,50} The production of **4** was initially confirmed by high-resolution mass spectroscopy (HR-MS), where a strong ionic peak at $m/z = 636.1735$ was observed (Supplementary Fig. 14), in comparison to that of 632.1438 found for **2**. When D_2 was employed for the reaction, HR-MS analysis of the reaction mixture revealed an ionic peak at 640.2057 (Supplementary Fig. 15). This finding can be rationalized by the addition of two molecules of D_2 to compound **2** through hydrodeuteration of the vinyl moiety and oxidative addition of D_2 to the molybdenum center. At room temperature, the hydride resonances of **4** coalesced into the baseline in the ^1H NMR spectrum.⁵¹ However, upon cooling the $\text{C}_6\text{D}_5\text{Cl}$ solution to 253 K, the hydride resonances appear as well-resolved peaks in the ^1H NMR spectrum. One hydride ligand shows a characteristic upfield signal at δ -4.19 as a doublet, while the signal of the other hydride is much further

downfield with a chemical shift of δ 5.16 (dd).⁵² The ^1H - ^1H COSY spectrum recorded at 253 K shows that the two sets of hydride signals correlate with each other with an extremely large $^2J_{\text{H-H}}$ of 110 Hz (Fig. 3e).⁵¹”

iv) The chemical shift difference of the two hydrido ligands in **4** is extremely high which makes one wonder whether the low-field signal is indeed a hydride. Did the authors consider other options such as an η^1 -bound H_2 ? At the very least, a possible explanation for the large difference should be discussed.

Response: We have recollected NMR spectra of **4** in $\text{C}_6\text{D}_5\text{Cl}$ at room temperature. The structure of two hydrido ligands was further examined by 2D ^1H - ^1H COSY spectroscopy. The literature indicates that two hydrido ligands are likely to occur in one positive and one negative position. (*Dalton Trans.* **44**, 18945–18956 (2015).) The DFT optimized structure of **4** with two hydrides is similar to the X-ray structure and the computed chemical shifts for these two hydrides are -3.7 and 2.9 ppm, respectively, which is consistent with the distinct chemical shifts observed experimentally. According to computation, the hydride closer to the P-donor is more deshielded, i.e., has the downfield signal. This hydride is in a highly congested spot. Other than soft of being in the deshielded region of a phenyl ring (not very close though, 2.04 Å from the closest ortho H of the phenyl ring) and where we would expect a lobe from the σ^* orbital of a P–C bond (P–H distance 2.34 Å), we have not found any unusual features of this hydride from the computational results. The Mulliken charge on both hydrides is negative (-0.30 and -0.18), with the downfield hydride being less negative. We have added the computed chemical shift to manuscript. If we move the downfield hydride toward the upfield hydride to make an η^2 - H_2 complex, the computed chemical shifts are -4.0 and -0.3 ppm, respectively.

v) X-ray data of **3** and **4** must be discussed by referring to literature data not only to the respective compounds in the manuscript.

Response: Thank you for pointing this out. We have added the following discussed on X-ray data of **3** and **4** by referring to literature data in our revised manuscript, see:

“The C–C bond lengths of the two η^2 - C_2H_2 ligands are 1.253(6) and 1.285(6) Å, which is consistent with the observation of two inequivalent side-on acetylene ligands in the NMR spectra. These C–C lengths are comparable to those found for the molybdenum acetylene complexes of $\text{Mo}(\text{C}_2\text{H}_2)(\text{dppe})_2$ (1.265(7) Å)⁴¹, $[\text{MoO}(\text{C}_2\text{H}_2)(6\text{-MePyS})_2]$ (1.2649(17) Å),¹⁵ and $\text{Mo}(\text{C}_2\text{H}_2)(\text{t-BuS})_2(\text{t-BuNC})_2$ (1.28(2) Å).¹³”

“In general, the coordination geometry of **4** is similar to those eight-coordinate

molybdenum analogs such as $[\text{CpMo}(\text{CO})(\text{PEtNMePEt})(\text{H})_2]^+$,²⁷ $[\text{Cp}^*\text{MoH}_2(\text{dppe})(\text{MeCN})]^+$,⁵¹ and $[\text{CpMoH}_2(\text{PMe}_3)_3]^+$.⁵⁰

vi) catalysis was performed with **2** only. I assume that **4** was not investigated for stability reasons. A comment must be provided.

Response: thanks for the queries. As the referee said, compound **4** is not stable in the solution, especially in the absence of H_2 . We provide a comment in the revised manuscript as:

“It is important to mention that complex **4** is unstable in solution and undergoes gradual degeneration, resulting in the formation of unidentified species.”

We also investigated the catalytic performance of complex **4** under the same catalytic conditions of **2**. We found that complex **4** also efficiently catalyzes hydrogenation of styrene:

A d_8 -THF solution of styrene (0.2 mmol), and **4** (4 μmol , 2 mol%) contained in a J. Young NMR tube, was degassed and then exposed to H_2 (1 atm) After reaction for 12 h at room temperature., the reaction produced ethylbenzene (**6a**) in 95% NMR yield.

Supplementary Figure 2. ^1H NMR (500 MHz, d_8 -THF) spectrum recorded for the hydrogenation of **5a** catalyzed by **4**.

vii) The comment on the tolerance of Mo catalyst on page 9 I cannot support. The cited literature refers to Mo(VI) from e.g. Schrock which are of course oxophilic while compound **2** in oxidation state +II is far less oxophilic. Thus, please rephrase.

Response: thanks for the careful review, we modified the statements to:

“However, olefins bearing various oxygen-containing functional groups, such as ether (**6t**, **6u**), ester (**6x**), amide (**6w**), and alcohol (**6y**, **6z**) all underwent smooth hydrogenation. This is noteworthy as such substrates were rarely tolerated by high-valent Mo-based catalysts.^{53,54}”

viii) Figure 6: is the structure of Int9 possibly wrong? I cannot see how a molybdenum carbene can be formed? It is the product of acetylene insertion which gives a 7-membered metallacycle with three C-C double bonds but no metal-carbon double bond. Please comment on it.

Response: The ring expansion of the 5-membered metallacycle into a 7-membered metallacycle is a step that has many outcomes based on the potential energy surface scans. The one we drew is the immediate product from the C-C bond formation reaction, i.e., π -allyl on one side and carbene on the other side of the metallacycle based on bond lengths; this initial intermediate is metastable and isomerizes into the 7-membered metallacycle as the review suggested with a free energy barrier of 0.5 kcal/mol. On paper, this 7-membered metallacycle should have three C-C double bonds; however, the bond lengths around the metallacycle seem to suggest that the C-M bonds being double bonds, which could be due to the significant pi-back donation from the metal center. We have updated the Scheme to reflect such a change; for simplicity, we removed the initial C-C formation product and draw the 7-membered metallacycle with a delocalized circle round the ring.

ix) Figure 6: it interesting that compound **3** is isolated in 59 % yield even though the barriers for polyacetylene formation are lower. Is **3** formed immediately or only after most acetylene has been converted to polyacetylene? Did the authors try to use stoichiometric amounts of acetylene?

Response: Thank you for the queries. Based on our calculations, the formation of **Int3** is the slow step. When a small fraction of **Int2** turns into **Int3**, the fast polymerization initiated by **Int3** depletes C_2H_2 quickly. The remainder of **Int2** will just turn into **3** after the C_2H_2 concentration drops below a certain threshold. Below is our explanation why

low C_2H_2 concentration favors the formation of **3**. From **Int3**, the polymerization route needs C_2H_2 to proceed, i.e., the rate will slow down significantly when the C_2H_2 concentration drops. However, the formation of **3** is intramolecular and does not get affected by this concentration drop. This hypothesis can be further demonstrated with our stoichiometric reaction, i.e., the reaction of **2** with 5 eq. of C_2H_2 monitored by ^{31}P NMR spectroscopy (see below). When the concentration of C_2H_2 is so low, the reaction gives 64% of **3**, 14% unreacted **2** and a few stalled intermediates (which might be **Int2**). When the polymerization reaction under 1 atm of C_2H_2 was monitored by ^{31}P NMR experiment, (see Figure SS1 added), the formation of **3** was observed in 20 minutes. It is worth noting that in this NMR tube, reaction the diffusion of C_2H_2 into solution is slow without stirring. Consequently, the initial C_2H_2 concentration is low, which favors the formation of **3**. As such, the immediate formation of **3** is expected and observed.

Under N_2 atmosphere, in an NMR tube fitted with a rubber stopper, **2** (15mg, 0.01 mmol) and 1,3,5-trimethoxybenzene (1.68 mg, 0.01 mmol) were dissolved in d_8 -THF (0.6 mL). The tube was taken out from the glovebox and injected C_2H_2 (1.23 mL, 0.05 mmol) using a syringe at $25^\circ C$ ($V_m = 24.5$ L/mol). After the indicated time, we monitored changes in ^{31}P NMR and 1H NMR spectra, respectively.

Figure SS1. $^{31}\text{P}\{^1\text{H}\}$ NMR (202 MHz, d_8 -THF) spectra depicting the reaction progress of **2** with C_2H_2 at 0 min, 10 min, 30 min and 60 min.

x) Figure 7: I find it surprising that between Int2A and Int3A no dihydride is found while later the dihydride is isolated (even 68 % yield). If no dihydride is formed, how is the alkyl formed? Do the authors consider heterolytic splitting of H_2 forming H^- and H^+ which protonates the vinyl? While this would be similar to reactions with acetylene, I doubt it. A dihydride seems the most likely TS. A comment on what TS1A might be, must be provided.

Response: Indeed, what we found based on the potential energy surface scan is the heterolytic splitting of the coordinated H_2 between the metal and the olefin. The following structural description of **TS1A** has been added to the discussion, “In **TS1A** the distance between the two hydrogen atoms originated from H_2 is 1.51 Å, shorter than that of a typical dihydride; **TS1A** is most consistent with the heterolytic splitting of a coordinated H_2 between the olefin carbon and the metal center, i.e., deprotonation of the H_2 ligand by the alkene ligand.”

xi) Page 2: scarce must be scarce.

Response: Thanks for the careful review. It has been changed to “scarce”.

xii) Since dissociation of pyridine the authors should consider using methyl pyridines or lutidines which could possibly speed up catalysis even more.

Response: We appreciate this reviewer for the insightful comments.

Similar to the reaction with pyridinium salts, the molybdenum acetylene compound also reacts with 2,6-lutidinium salts. According to ^{31}P NMR (Supplementary Fig. 7), the reaction also produced the analogue vinyl complex **2'**.

Under the identical catalytic conditions of **2**, the catalytic performances of **2'** in the hydrogenation of styrene were compared. (please see Supplementary Fig. 8 and Fig. 9) Synthesis of complexes **2** and **2'**: Under N_2 atmosphere, a green solution of **1-C₂H₂** (30 mg, 0.054 mmol) in 3 mL of THF was cooled to -20 °C. The resulting solution was treated with 2,6-lutidinium salts (0.053 mmol) in 2 mL of THF, which caused a change in the solution color from green to brown. After the removal of the solvent under vacuum, the solid was washed with hexane (5-10 mL). The corresponding complexes **2** and **2'** were precipitated and separately isolated as brown powder.

Hydrogenation of **5a** catalyzed by **2** and **2'**: In an N_2 -filled glovebox, to three J. Young NMR tubes charged separately with catalyst **2** and **2'** (0.004 mmol), **5a** (0.2 mmol) in d_8 -THF (0.6 mL), was added 1,3,5-trimethoxybenzene (11.2 mg, 0.067 mmol) as the internal standard. The tube was taken out from the glovebox and immersed in a liquid nitrogen bath, and gently degassed under a vacuum. The solution was then warmed up to room temperature and pressurized with H_2 gas (1 atm). After reaction at room temperature for 3 h, the solution was analyzed by ^1H NMR to determine the yield of **6a**.

Supplementary Figure 7. $^{31}\text{P}\{^1\text{H}\}$ NMR spectrum of **2** and **2'**.

Supplementary Figure 8. ^1H NMR (500 MHz, d_8 -THF) spectrum recorded for the hydrogenation of **5a** catalyzed by **2** for 3h.

Supplementary Figure 9. ¹H NMR (500 MHz, *d*₈-THF) spectrum recorded for the hydrogenation of **5a** catalyzed by **2'** for 3h.

Reviewer #2 (Remarks to the Author):

The manuscript by Minghui Xue and coworkers reports a molybdenum-sulfur complex **2** that catalyzes acetylene polymerization (with substantial catalyst deactivation to **3**, and potentially to other species) and catalyzes alkene hydrogenation through a Mo(IV) dihydride **4**. The manuscript extends recent work by the Beller and Chirik groups and others, all of which was published in specialized chemistry journals (refs 31-37). DFT-predicted mechanisms for reaction of **2** with H₂ and with ethylene include rather high barriers to catalyst activation (Int2 to Int3, barrier 23.5 kcal/mol, TS3A, barrier 23.7 kcal/mol). The authors also include a speculative mechanism for alkene hydrogenation based on the DFT-predicted reaction of **2** with H₂. A detailed DFT mechanism would be a better methodology. It is not clear, from the authors' writeup, that this Mo-based alkene hydrogenation catalyst will outperform existing non-precious-metal alkene hydrogenation catalysts (see the 2015 review by Chirik). In my judgement, the manuscript has a modest significance as a follow-up to refs 31-37. It would be more appropriate as a submission to the specialized chemistry journals that published refs 31-37.

Response: The main source of error for the computed energetics is the imperfect solvation, especially when the catalyst is cationic and when the reaction involves charge redistribution. Our catalyst is cationic and both steps in question involve significant charge redistribution, i.e., oxidative cycloaddition from **Int2** to **Int3** and reductive elimination from **Int3A** to **Int4A** (even the step from **Int2A** to **Int3A** involves charge redistribution). It is well known that the solvation issue is significant for this kind of systems; so, a certain degree of errors in computed free energy is expected. In addition, the barriers of 23.5 kcal mol⁻¹ (from **Int2** to **Int3**) is also slightly inflated because the default standard state in Gaussian for gaseous starting material is p⁰ = 1, but the reaction we are considering is in solution, i.e., the standard state should be [X]₀ = 1 M instead. Such discrepancy gives rise to ~2 kcal/mol energy difference, i.e., 23.5 vs 21.5 kcal/mol.

Reviewer #3 (Remarks to the Author):

The submitted work by Wenguang Wang and coworkers is not eligible for publication in Nature Communications. The paper deals with the hydrogenation of olefins facilitated by a Mo complex. The reduction of alkenes with gaseous hydrogen is a well-established catalytic transformation and Mo is well-known for its ability to catalyze redox processes. Furthermore, the substrate scope is rather narrow and the authors merely report NMR yields for their products. Hence, the given manuscript lacks the novelty which is necessary for publication in Nat. Commun. However, the experiments including the calculations and the NMR studies were all well-performed and thus I recommend submission of the subject manuscript to a more catalysis-dedicated journal such as ChemCatChem.

Response: Thanks for acknowledging the technical quality of our work. To further demonstrate the capability of our catalyst, we have expanded the substrate scope and performed two large-scale late-stage hydrogenation of a natural product with isolated yields.

Reviewer #4 (Remarks to the Author):

This manuscript reported a molybdenum complex for catalytic hydrogenation of various aromatic and non-activated aliphatic alkenes. As author reported, after protonated, 1-C₂H₂ converted to complex 2, which is the pre-catalyst for various alkenes hydrogenations. This complex can react with H₂ and form Mo(IV)-dihydride 4 which is related to the proposed mechanism. Various characteristic methods as well as

DFT calculations were used for the compounds' characterizations, reactions analysis and mechanism study. This is a great topic and complex 2 is an intriguing hydrogenation catalyst; however, to be suitable for publication in Nature Communications, the manuscript needs to be improved, in particular, by addressing the following key aspects:

Response: We appreciate this reviewer for the positive comments and recommendation of our work.

(1)The quality of characterization of Mo(IV)-dihydride **4** needs to be improved. The ¹H NMR spectrum looks not pure. For SXRD, with alert A errors, and the R factor of 0.09, the H cannot be located accurately. Please provide better crystallography data or other evidence. Authors may also try deuterium experiment to locate the hydrides. Besides, the authors assigns the two hydrides, one in upfield and the other in downfield, further references and characterizations need to be provided. Moreover, for all four molybdenum complexes, please identify the solvent residue signal, water peak, and impurities in ¹H-NMR. For ³¹P-NMR, what is the internal or external standard? For ESI-MS, please also add the whole scope of the spectra first, and then give the zoom in of founded peaks.

Response: We agree with the reviewer's suggestion.

(i) For SXRD, we have corrected the alert A error, and the R factor of 0.08.

(ii) At room temperature, the hydride resonances of **4** coalesced into the baseline in the ¹H NMR spectrum. We have recollected NMR spectra of **4** in C₆D₅Cl at 253K. The structure of two hydrido ligands was further examined by 2D ¹H-¹H COSY spectroscopy. The literature indicates that two hydrido ligands are likely to occur in one positive and one negative position. (*Dalton Trans.* **44**, 18945–18956 (2015).)

The attempts to locate the hydrides by deuterium experiment at 253K are always unsuccessful because compound **4** is not stable in the solution. Especially, the NMR deuterium spectrum exhibits a decrease in the signal-to-noise ratio and a weakening of the signal under 253K.

In addition, The DFT optimized structure of **4** with two hydrides is similar to the X-ray structure and the computed chemical shifts for these two hydrides are -3.7 and 2.9 ppm, respectively, which is consistent with the distinct chemical shifts observed experimentally. According to computation, the hydride closer to the P-donor is more deshielded, i.e., has the downfield signal. This hydride is in a highly congested spot. Other than sort of being in the deshielded region of a phenyl ring (not very close though, 2.04 Å from the closest ortho H of the phenyl ring) and where we would expect a lobe from the σ* orbital of a P–C bond (P–H distance 2.34 Å), we have not found any unusual features of this hydride from the computational results. The Mullikan charge

on both hydrides are negative (-0.30 and -0.18), with the downfield hydride being less negative. We have added the computed chemical shift to manuscript to support our assignment.

(iii) There is no internal or external standard in the ^{31}P NMR spectrum.

(iv) We performed the whole scope of spectra in ESI-MS.

Supplementary Figure 11. ESI-MS spectrum of 1-C₂H₂.

Supplementary Figure 12. ESI-MS spectrum of 2.

Supplementary Figure 13. ESI-MS spectrum of **3**.

Supplementary Figure 14. ESI-MS spectrum of **4**.

(2) In catalytic hydrogenation section, the authors may check the Hammett σ parameters of the substituents, the yields percentage may be related to the electron density. The effects (such as electronic/steric) on alkenes' yields need to be further analyzed to associate the mechanism. The hydrogenation reaction were only studied by small scale in NMR tube. Did the authors try bulk catalysis and isolate the products? Bulk catalysis experiment can further test the catalytic ability of the Mo complex.

Response: We appreciate this reviewer for the insightful comments.

i) With para-substituted alkenes, a linear relationship was further observed through a Hammett plot. (see the Supplementary Table 2 and Fig. 10)

Supplementary Table 2. Data for Hammett-plot.

Entry	Substituent X	σ	k_X	k_X/k_H	$\log(k_X/k_H)$
1	p-MeO	-0.27	3.88×10^{-4}	1.23	0.09
2	p-Me	-0.17	3.46×10^{-4}	1.10	0.04

3	p-H	0	3.15×10^{-4}	1	0
4	p-F	0.06	2.67×10^{-4}	0.85	-0.07
5	p-OCOMe	0.31	1.60×10^{-4}	0.51	-0.29
6	p-COOMe	0.45	1.45×10^{-4}	0.46	-0.34
7	p-CF ₃	0.54	3.89×10^{-4}	0.40	-0.39

Hammett equation in catalytic hydrogenation:

$$\log\left(\frac{k_X}{k_H}\right) = -0.64\sigma$$

Supplementary Figure 10. Hammett-plot for hydrogenation.

The related results have been added into the revised manuscript, please see:

“To evaluate the influence of electronic variation in the substituent on the reaction rate, kinetic studies were performed on hydrogenation of the para-substituted styrene derivatives (p-X-styrene, **5a-5g**). The reaction progress was monitored using ¹H NMR spectroscopy, revealing a linear reaction profile within the initial 4 hours (Supplementary Fig. 10). The reaction rate was found to strongly depend on the electronic nature of the para-substituent. The kinetic data are correlated with the standard Hammett σ_{para} values,⁵⁵ resulting in a negative slope of $\rho = -0.64$ (Fig. 3g). The small absolute value of the Hammett electronic parameter suggests that the reaction site in the turnover-limiting transition state is more remote than the benzylic carbon from the para-X-group, ruling out the insertion step being the turnover-limiting.⁵⁶ The small negative ρ is consistent with the reductive elimination step being turnover-limiting, where the terminal carbon of the styrene substrate is involved.”

ii) We have added bulk catalytic hydrogenation of Allylestrenol (**5ad**) with general reaction condition. The isolated product **6ad** and *d*₂-**6ad** were identified by ¹H NMR

and ^2H NMR. (see Supplementary Fig. 59 and Fig. 60).

In an N_2 -filled glovebox, a flame-dried Schlenk tube was charged with catalyst **2** (149.6 mg, 2 mol%), **5ad** (601.0 mg, 2.0 mmol) in d_8 -THF (3.0 mL). The tube was taken out from the glovebox and immersed in a liquid nitrogen bath, and gently degassed under vacuum. The solution was then warmed up to room temperature and pressurized with H_2 gas (1 atm). After reaction at room temperature for 12 h, the mixture was evaporated to dryness, and the crude product was purified by column chromatography on silica gel eluting with petroleum ether/EtOAc (10: 1) to give the corresponding product **6ad** (white solid, 540.6 mg, 90%).

Following the procedure of reaction with H_2 , **6ad** also reacts with D_2 . After the reaction, the deuterated product d_2 -**6ad** (white solid, 98.3 mg, 91%) was obtained.

(3) In the hydrogenation mechanism cycle (Fig 8), besides Mo(IV)-dihydride, is there any other intermediates observed or confirmed by any wet experiments?

Response: There are no other intermediates were observed in catalytic hydrogenation by wet ^{31}P NMR. The X-ray structure and NMR spectrum of complex **4** indicate that the cleavage of H_2 at the molybdenum center without intermediate. The related examples of oxidative addition of H_2 at Mo center are reported. (*J. Am. Chem. Soc.* **130**, 16187–16189 (2008); *Eur. J. Inorg. Chem.* **2011**, 141–149 (2011).)

(4) The structure of the main-text needs to be improved. In discussion part, there are too much DFT details. Conclusions of all important experiments and future perspective need be re-written in better way.

Response: We have re-written this part based on the suggestions from this reviewer and

reviewer 1.

Regarding the conclusions and future perspectives, we reorganized the conclusion parts:

“We have demonstrated the protonation of a molybdenum(II)-acetylene complex using a pyridinium salt, resulting in the formation of a novel cationic Mo(II)-vinylthioether complex. This complex exhibits efficient catalytic activity for hydrogenation of various aromatic and aliphatic alkenes at ambient temperature. The reaction exhibits excellent substrate compatibility and high functional group tolerance without the need for any additives. Mechanism studies reveal that the catalysis involves a Mo(IV) dihydride intermediate arising from an uncommon oxidative addition of H₂ to the cationic Mo(II) center. Additionally, we found that complex **2** displayed intriguing reactivity toward C₂H₂, causing the vinylthioether ligand to undergo chain propagation and yield a dangling triene chain. Our future studies will focus on modifying the coordination sphere of [Cp*Mo(1,2-Ph₂PC₆H₄SR)] to enhance the catalytic performance in the transformation of C₂H₂ and substituted alkynes.”

Minor Questions/Suggestions:

(1) Please add percentage yields in Fig.2., and add the temperature in the step from **1**-C₂H₂ to **2**.

Response: Thanks for the careful review. In the revised Fig.2, the yields have been provided for each step, and the reaction temperature has been noted for the conversion of **1**-C₂H₂ to **2**.

(2) Merge Figure 3 and 4 together or move figure 4 into Supplementary Information.

Response: Figure **3** and **4** has been combined, please see Figure 3 in the revised manuscript.

Reviewers' Comments:

Reviewer #1:

Remarks to the Author:

All my concerns have been adequately addressed and corrections were made to my full satisfaction.

I particularly appreciate the experiment with 2,6-lutidinium BArF which reveals a slightly higher yield in styrene reduction. However, this experiment should also be mentioned in the main text of the manuscript and not just in the supporting information.

After that, I fully recommend publication of the manuscript in Nature Communications.

Reviewer #2:

Remarks to the Author:

The authors' response on solvation models is well-taken, but does not change my judgement of the manuscript as a whole. In my judgement, the manuscript has a modest significance as a follow-up to refs 31-37. It would be more appropriate as a submission to the specialized chemistry journals that published refs 31- 37.

Reviewer #4:

Remarks to the Author:

Thanks for the authors' reply and updated manuscript. However, there are still three main questions in the current manuscript.

1. For the X-ray structure of complex 4, the authors fix the R factor from 0.09 to 0.08, however, 0.08 is still not enough for identifying hydrogen atoms. For publication, it should usually be below 0.05, and R=0.08 cannot locate hydrogen accurately, thus it cannot prove the existence of hydrides in the X-ray technique.

2. To obtain the accurate P-NMRs, I suggest the authors to retest all P-NMR with inside or outside standards.

3. the whole mass-spec, it seems that the products are not pure, please identify the other peaks in mass. The author claimed that there were no other intermediates or by-products, however, from the mass-spec of complex 4, it is not pure, it seems that there are other intermediates.

I think if the authors can answer the above-mentioned questions and update the experimental data, the manuscript will be improved a lot.

Response to Reviews: NCOMMS-23-32745A

Reviewer #1 (Remarks to the Author):

All my concerns have been adequately addressed and corrections were made to my full satisfaction.

I particularly appreciate the experiment with 2,6-lutidinium BARF which reveals a slightly higher yield in styrene reduction. However, this experiment should also be mentioned in the main text of the manuscript and not just in the supporting information. After that, I fully recommend publication of the manuscript in Nature Communications.

Response: we thank this reviewer for the positive comments and for supporting the publication of our work in Nature Communication. The results have been discussed in the main text, please see:

“Additionally, experimental studies have revealed that an excess of pyridine impedes the conversion of **2** to **4** and the catalytic hydrogenation of styrene. These findings correspond with the DFT results, indicating the importance of pyridine dissociation in the hydrogenation processes (Supplementary Fig. 5 and Fig. 6). In particular, when employing $[\text{Cp}^*\text{Mo}(\text{1,2-Ph}_2\text{PC}_6\text{H}_4\text{S-CH=CH}_2)(\text{2,6-lutidine})]\text{BARF}_4$, which is *in-situ* generated by **1**-C₂H₂ with 2,6-lutidinium salts, as the catalyst, the hydrogenation of styrene with H₂ at room temperature for 3 hours results in a slightly higher yield of **6a** (35%) in comparison to the 26% yield obtained in the styrene reduction catalyzed by **2** under identical reaction conditions (Supplementary Fig. 7- Fig. 9).”

Reviewer #2 (Remarks to the Author):

The authors' response on solvation models is well-taken, but does not change my judgement of the manuscript as a whole. In my judgement, the manuscript has a modest significance as a follow-up to refs 31-37. It would be more appropriate as a submission to the specialized chemistry journals that published refs 31- 37.

Response: we appreciate the referee for the valuable time spent providing critical comments during the review process.

Reviewer #4 (Remarks to the Author):

Thanks for the authors' reply and updated manuscript. However, there are still three main questions in the current manuscript.

1. For the X-ray structure of complex 4, the authors fix the R factor from 0.09 to 0.08, however, 0.08 is still not enough for identifying hydrogen atoms. For publication, it should usually be below 0.05, and $R=0.08$ cannot locate hydrogen accurately, thus it cannot prove the existence of hydrides in the X-ray technique.

Response: we deeply appreciate the insightful comments provided by this reviewer, which have significantly contributed to the improvement and shaping of this work for publication in Nature Communications.

We agree that X-ray diffraction cannot locate hydrogen with high accuracy, which is reflected by the high standard deviations of the bond lengths and angles involving H atoms in general (even for structures with 2% R-factors). However, we are not trying to use X-ray crystallography to claim the accuracy of the hydride positions in this work. Our NMR data, DFT calculation results (both the optimized structure and the computed chemical shifts of the hydrides), MS data, and the X-ray crystallographic data are all consistent with the assigned structure.

In the revised manuscript, we have modified the descriptions about the Mo-H distances.

2. To obtain the accurate P-NMRs, I suggest the authors to retest all P-NMR with inside or outside standards.

Response: Thank you for the suggestions. We have recollected the ^{31}P NMR spectra of complex 1-4 by adding a capillary with 4% H_3PO_4 as internal standard.

Supplementary Figure 17. $^{31}\text{P}\{^1\text{H}\}$ NMR (202 MHz, d_8 -THF) spectrum of 1-C₂H₂ with and without 4% H_3PO_4 used as internal standard in the capillary at 273 K.

Supplementary Figure 20. $^{31}\text{P}\{^1\text{H}\}$ NMR (202 MHz, d_8 -THF) spectrum of **2** with and without 4% H_3PO_4 used as internal standard in the capillary.

Supplementary Figure 23. $^{31}\text{P}\{^1\text{H}\}$ NMR (202 MHz, d_8 -THF) spectrum of **3** with and without 4% H_3PO_4 used as internal standard in the capillary.

Supplementary Figure 26. $^{31}\text{P}\{^1\text{H}\}$ NMR (202 MHz, d_8 -THF) spectrum of **4** with and without 4% H_3PO_4 used as internal standard in the capillary at 253 K.

3. the whole mass-spec, it seems that the products are not pure, please identify the other peaks in mass. The author claimed that there were no other intermediates or by-products, however, from the mass-spec of complex **4**, it is not pure, it seems that there are other intermediates. I think if the authors can answer the above-mentioned questions and update the experimental data, the manuscript will be improved a lot.

Response: thank you for the queries. By analyzing the mass spectra, we can see that the predominant peaks correspond to the (ionic) fragments resulting from the dissociation of ligands such as C_2H_2 , pyridine, and hydrides. Please see below:

Supplementary Figure 11. HRMS spectrum of $1\text{-C}_2\text{H}_2$.

Supplementary Figure 12. HRMS spectrum of **2**. *Note:* the ionic peak at 553.1019 corresponds to $[\text{Cp}^*\text{Mo}(1,2\text{-Ph}_2\text{PC}_6\text{H}_4\text{S}-\text{CH}=\text{CH}_2)]^+$ resulting from **2** with the loss of a pyridyl fragment.

Supplementary Figure 13. HRMS spectrum of **3**. *Note:* the ionic peak at 631.1431 corresponds to the species resulting from **3** with the loss of a C₂H₂ fragment.

Supplementary Figure 14. HRMS spectrum of **4**. *Note:* the ionic peak at 555.1166 corresponds to $[\text{Cp}^*\text{Mo}(1,2\text{-Ph}_2\text{PC}_6\text{H}_4\text{S}-\text{CH}_2\text{CH}_3)]^+$ resulting from **4** with the loss of a pyridyl fragment and two hydride ligands, while the peak at 323.1015 corresponds to the phosphine moiety Ph₂PC₆H₄SCH₂CH₃.

Supplementary Figure 15. HRMS spectrum of d_4 -4. *Note:* the ionic peak at 557.5936 corresponds to $[\text{Cp}^*\text{Mo}(1,2\text{-Ph}_2\text{PC}_6\text{H}_4\text{S-CHDCH}_2\text{D})]^+$ resulting from **4** with the loss of a pyridyl fragment and two deuterium ligands.